# Frequency Distribution Model of Wind Speed Based on the Exponential Polynomial for Wind Farms

**Lingzhi Wang [1,2,3,\*], Jun Liu [2] and Fucai Qian [2,4]**

[1]   School of Automation, Xi'an University of Posts and Telecommunications, Xi'an 710121, China
[2]   School of Automation and Information, Xi'an University of Technology, Xi'an 710048, China;
     liujun0310@sina.com (J.L.); fcqian@xaut.edu.cn (F.Q.)
[3]   Shaanxi Key Laboratory of Complex System Control and Intelligent Information, Xi'an 710048, China
[4]   Autonomous Systems and Intelligent Control International Joint Research Center, Xi'an Technological
     University, Xi'an 710021, China
[\*]   Correspondence: wlz@xupt.edu.cn; Tel.: +86-137-2063-6337

**Abstract:** This study introduces and analyses existing models of wind speed frequency distribution in wind farms, such as the Weibull distribution model, the Rayleigh distribution model, and the lognormal distribution model. Inspired by the shortcomings of these models, we propose a distribution model based on an exponential polynomial, which can describe the actual wind speed frequency distribution. The fitting error of other common distribution models is too large at zero or low wind speeds. The proposed model can solve this problem. The exponential polynomial distribution model can fit multimodal distribution wind speed data as well as unimodal distribution wind speed data. We used the linear-least-squares method to acquire the parameters for the distribution model. Finally, we carried out contrast simulation experiments to validate the effectiveness and advantages of the proposed distribution model.

**Keywords:** wind farms; wind speed frequency distribution; exponential polynomial model; linear-least-squares method

## 1. Introduction

Investment in renewable energy sources, including wind power plants, is of particular importance because of the increased efficiency of clean energy, and the need to reduce pollution and fuel consumption [1]. As wind generation technologies improve, this form of energy production becomes a valuable alternative to conventional energy sources [2]. The proportion of energy generated by wind is increasing due to recent technology and efficiency improvements, as well as government funding [3]. An important problem in using wind power is their uncertain nature and characteristic of being unforeseen [4]. To develop and utilise wind energy resources efficiently, the characteristics of wind energy resources first need to be analysed and studied [5]. The assessment of energy resources at wind farms is the foundation for development. Discovering the characteristics of wind speed frequency distribution in wind farms is the key to the research of wind energy resources. The wind speed frequency distribution refers to the probability density function of wind speed, which describes the complete statistical properties of wind speeds displaying random behaviour [6].

The different descriptions of wind speed frequency distribution for wind farms directly reflects the different conditions of wind energy resources at a site. Its rationality and accuracy will have a direct influence on the final decisions of wind turbine selection, power generation estimation and economic benefit evaluation of wind farms. There remain critical differences between the actual and designed power generation of many wind farms with regards to the practical operation of the wind

farm and its evaluation analysis after operation. One of the main reasons for this discrepancy is that there is a significant error in the description of wind energy resources for wind farms [7]. It is therefore of great significance to study wind speed frequency distribution models for better development of wind farms.

Many scholars have studied and put forward models to fit the observed frequency distribution of wind speed. Stewart proposed that Weibull distribution is very adaptable to the frequency distribution of different shapes [8]. Hafzullah Aksoyv [9] and Azami Zaharim [10] thought that wind speed frequency distribution can be fitted by either normal distribution, Weibull distribution, or the autoregressive model, and put forward relevant adaptive conditions, but the error is much larger than the actual distribution. Stanton E. Tuller [11] and others described the wind speed characteristics of common, mixed and three-parameter Weibull distributions, but there are many problems emerging in fitting at zero and low wind speed. Ahmad [12] thought that Weibull distribution cannot fit the case of zero wind speed, so revised it and presented a new model, but it is fundamentally hard to solve the problem using this means. Pishgar-Komleh [13] and others applied Weibull and Rayleigh distribution functions to find out the best fitting tool to the wind speed data, but it can be observed from results that at some wind speeds there are significant errors between the models and the observed wind speed data. Soulouknga [14] et al. used Weibull distribution to make analysis of wind speed data and wind energy potential and found that the Weibull shape parameter and scale parameter increase with altitude. Vladislovas [15] and his collaborators investigated wind power density distribution at locations with low and high wind speeds using the Weibull model and provided four numerical methods for evaluating Weibull parameters. Asghar and Liu [16] proposed a hybrid intelligent learning based adaptive neuro-fuzzy inference system to accurately estimate Weibull wind speed probability density function.

All the above studies are based on the Weibull distribution model. The following studies have proposed distributions other than the Weibull distribution model.

Kostas [17] et al. put forward the gamma probability distribution function to replace Weibull distribution for the area under study, but it still failed to achieve the desired results. Calf [18] et al. discovered that the wind speed frequency distribution usually covers three kinds of situations and made use of the Dirichlet function to fit, but this function is quite complex. Loukatou [19] proposed and tested an Ornstein-Uhlenbeck geometric Brownian motion model over continuous time to represent the wind speed, avoiding the problems of using the Weibull distribution model. Elfarra and Kaya [20] designed a novel way to define the probability density for wind speed data using splines and validated that spline-based probability density functions produce a minimum fitting error for all the analysed cases.

Therefore, many models for wind speed exist in the literature, but they all lack the accuracy to fit measured data at zero and low wind speed.

Inspired by the above work, and aiming at the problem of a poor fit between the selected model and measured data, especially the problem that the probability density is not zero at zero wind speed and that there is a big gap between the theoretical calculation and measured data at low wind speed, in this paper we propose an exponential polynomial model to describe and calculate wind speed frequency distribution. The fitting effect is verified by comparison experiments based on the measured data. The exponential polynomial can be transformed into a linear equation set with respect to parameters, therefore, we adopt the linear-least-squares method to achieve the solution for the exponential polynomial model.

In some cases, because the measured wind speed data has more than one mode in the probability density, the conventional distributions, including Weibull, fail to fit the wind speed data. This highly affects the technical and economic assessment of a wind energy project by causing crucial errors. To address this problem, Elfarra and Kaya [20] made use of splines to define the probability density for multimodal wind speed data and proved the validity of the method. In fact, the exponential polynomial model presented in this paper can also fit the frequency distribution of multimodal wind

speed. Meanwhile, there are differences in three main points from the literature [20]: (1) The piecewise cubic polynomial is used for constructing a spline in the literature. When optimizing spline coefficients the values of three functions need to be minimised, including the values of function at each node, and its first derivative values at the first and last nodes. This makes the optimization problem much more complex than that described in this paper. We do not need to calculate and minimise the first derivative value, so it is simpler and easier to deal with. (2) From the literature, obtaining the optimum splines requires the solution of a constrained optimization problem with five constraints, therefore computation involving a lot of mathematical operations is necessary. The optimization problem based on the proposed model has no constraints, so the amount of calculation is small. (3) The parameters in the literature need to be initialised, while there is no need to set the initial value for the parameters in the linear-least-squares method in this paper.

The three main contributions of this paper compared to past work are summarised as follows:

(1)　The proposed exponential polynomial model is utilised as a novel method for modelling the frequency distribution of wind speed. Our idea provides an effective strategy for fitting the model to the observed frequency distribution at zero and low wind speed, better describing the actual distribution of wind energy resources, and making up for the missing piece in the field. Moreover, this work offers an analytical basis for the development of wind energy resources and is helpful for wind farm construction.

(2)　Although numerous approaches to solve parameters in the wind speed frequency distribution model exist in the literature, we adopt the linear-least-squares method because of the special form of the exponential polynomial model. The optimization algorithm is simple and requires very little computation. The order of polynomial can be changed flexibly according to demand, so that the fitting effect can be easily improved.

(3)　The exponential polynomial distribution model can describe not only the frequency distribution of unimodal wind speed, but also the frequency distribution of multimodal wind speed, thus more accurately assessing wind energy resources for wind farms.

The remainder of this paper is organised as follows. Following the introduction in this section, we introduce several typical distribution models for wind speed frequency distribution in Section 2. In Section 3 we propose the exponential polynomial distribution model and offer a technique based on the linear-least-squares method for solving parameters in the proposed distribution model. We undertake the description of simulation experiments in Section 4 and show simulation results. In Section 5, the results are analysed and discussed, demonstrating the advantage of the developed distribution model. Finally, we conclude the paper in Section 6.

## 2. Frequency Distribution Models of Wind Speed

Wind speed frequency defines the frequency of wind speed arising in each designated interval and can describe the conditions of wind energy resources at wind farm sites. It is an important parameter index in wind energy resources assessment and wind farm design. According to the measured wind speed, the formula for calculating wind speed frequency [21] is:

$$f(v_i) = \frac{i}{n} \times 100\% \tag{1}$$

where $n$ is the number of wind speed series in the observation period, $i$ is the number of wind speed series in the wind speed interval, and $v_i$ is the $i$th wind speed section.

Wind farms are generally built in places with relatively rich wind resources such as plains, coastal areas, and inland mountains. With different climates and geographical conditions at wind farm sites, wind speed and wind speed frequency distribution parameters are random. Wind speeds vary over time, creating a speed–time correlation. Therefore, the frequency distribution of wind speed can

be statistically analysed and processed according to the measured wind speed based on increments of time.

Because of the variety of wind speed characteristics and different forms of wind speed distribution, multiple frequency distribution models of wind speed can be used to fit the distribution of wind energy resources. There are many models to describe the characteristics of wind speed frequency distribution which can be used to predict the wind speed frequency distribution over each month. At present, the commonly used models are the Weibull distribution model, the Rayleigh distribution model, and the log-normal distribution model.

### 2.1. The Weibull Distribution Model

The Weibull distribution model is the most classical model used to fit wind speed frequency distribution [5,8–16]. The model has a strong adaptability to different frequency distribution and can well describe wind speed distribution, especially when estimating wind speed frequency distribution. It mainly includes the two-parameters Weibull distribution model and the three-parameters Weibull distribution model.

The three-parameters Weibull model can generally describe the distribution of wind energy resources. Its probability density function is as follows:

$$f_w(v) = \frac{k}{c}(\frac{v-\gamma}{c})^{k-1} \exp[-(\frac{v-\gamma}{c})^k] \tag{2}$$

where $k$ is the shape parameter, $1 < k < 3$, $c$ is the scale parameter, and $\gamma$ is the location parameter.

When $\gamma = 0$ is applied, model (2) can be simplified as a two-parameters Weibull distribution model. Because of its simple form and convenient calculation, it is widely used in engineering. Its probability density function is:

$$f_w(v) = \frac{k}{c}(\frac{v}{c})^{k-1} \exp[-(\frac{v}{c})^k] \tag{3}$$

The shape parameter $k$ determines the shape of the distribution curve. When $0 < k < 1$, $f(v)$ is a subtractive function about the wind speed; when $k = 1$, the distribution is of exponential type; when $k = 2$, it is called Rayleigh distribution; and when $k = 3.5$, Weibull distribution is very close to normal distribution. The larger the shape parameter $k$, the smaller the wind speed fluctuation. For very violent winds, such as polar winds, the shape parameter values are generally very small. When $c = 1$, it is called the standard Weibull distribution. The scale parameter $c$ represents the time characteristics of the wind speed and a specific correlation between wind speed distribution and average wind speed.

### 2.2. The Rayleigh Distribution Model

When $k = 2$ in the Weibull distribution model, it yields the Rayleigh distribution model, and its distribution function of wind speed frequency is:

$$f_R(v) = \frac{\pi}{2}\frac{v}{v_m^2} \exp[-\frac{\pi}{4}(\frac{v}{v_m})^2] \tag{4}$$

where $v_m$ is the mean wind speed over a certain period of time, with the calculation formula:

$$v_m = \int_0^\infty vf(v)dv \tag{5}$$

Combining with formula (3), we get:

$$v_m = c\Gamma(1 + \frac{1}{k}) \tag{6}$$

Hence, using the Rayleigh distribution model, if $v_m$ is known, the wind speed frequency distribution can be obtained.

### 2.3. The Log-Normal Distribution Model

In the initial stage of studying wind speed frequency distribution, the log-normal distribution model is usually used to fit wind speed frequency, and the function is:

$$f_N(v) = \frac{1}{v\sigma\sqrt{2\pi}}\exp[\frac{-(\ln v - \mu)^2}{2\sigma^2}] \tag{7}$$

where $\sigma$ is the shape parameter, and $\mu$ is the scale parameter. The calculation formulas of $\sigma$ and $\mu$ are respectively:

$$\mu = \frac{1}{n}\sum_{i=1}^{n}\ln v_i \tag{8}$$

$$\sigma^2 = \frac{1}{n-1}\sum_{i=1}^{n}(\ln v_i - \mu)^2 \tag{9}$$

## 3. The Exponential Polynomial Distribution Model

Some researchers have investigated Weibull distribution more thoroughly. The fit of results for the Weibull distribution are very good for the middle and high wind speed sections. However, there is a big gap between the theoretical calculation and measured data for the low wind speed section, especially in the zero-wind speed section. For example, the probability density of calculating zero wind speed is zero using Weibull's two-parameters model, but the measured results in many areas are not zero (the probability of actual zero wind speed in Erguna Banner of Inner Mongolia is 24%) [22]. Rayleigh distribution is a simplified model of Weibull distribution, so it also has the same deficiency.

To overcome the shortcomings of the above models, in this paper we try to propose an exponential polynomial model to describe the frequency distribution of wind speed. The mathematical description is as follows:

$$p(v) = C\exp(\sum_{i=1}^{n} a_i v^i) \tag{10}$$

where $C$ is the normalised constant, and $n$ is the highest order of exponential polynomial. When $i = 2$, it is a second order exponential polynomial model; when $i = 3$, it is a third order exponential polynomial model, and so on. The constant $a_i$ is determined through a parameter estimation method according to the measured wind speed distribution probability.

It is noticeable that model (10) does not equal zero when $v = 0$; this solves the problem that the probability density is not zero for zero wind speed. Model (10) can therefore be used to represent the frequency distribution of wind speed.

### 3.1. The Solution Algorithm Based on the Linear-Least-Squares Method

The wind speed frequency probability distribution parameters are important index parameters to characterise the statistical characteristics of wind energy resources and are also important and necessarily known parameters for wind farm planning [15,22–25].

In order to obtain the optimal parameters of the wind frequency distribution model, the performance index function is designed as:

$$J = \sum_{i=a}^{b}[p_m(v_i) - p(v_i)]^2 \tag{11}$$

In formula (11), *a* and *b* are respectively the minimum and maximum of the average wind speed over a different period of time, $p_m(v_i)$ is the probability calculated by the wind speed frequency distribution model and $p(v_i)$ is the measured wind speed probability when the wind speed is $v_i$ m/s.

According to Equation (10), the following polynomial is obtained:

$$\sum_{i=1}^{n} a_i v^i = \ln \frac{p(v)}{C} \qquad (12)$$

When *v* in Equation (12) is fixed by using sample points, Equation (12) becomes a linear equation with respect to $a_i$, so it can be solved through the linear-least-squares method. If we collect $N+1$ points $v_{i+1}$ from measured data, then the above formula will generate the following equation set:

$$\begin{cases} \sum_{i=1}^{n} a_i v_0{}^i = \ln \frac{p(v_0)}{C} \\ \sum_{i=1}^{n} a_i v_1{}^i = \ln \frac{p(v_1)}{C} \\ \qquad \vdots \\ \sum_{i=1}^{n} a_i v_N{}^i = \ln \frac{p(v_N)}{C} \end{cases} \qquad (13)$$

Thus we can acquire the solutions of $a_i$ using the linear-least-squares method. The linear-least-squares method is simple and has an obvious computational advantage. When optimizing parameters, there is also no need to set the initial value for the parameters in the linear-least-squares method [26].

Here it needs to be noticed that when solving the equation set (13) for the parameters with the linear-least-squares method, the number of data points selected must be more than that of the parameters, that is, $N > n$. Otherwise, there is no solution.

### 3.2. The Algorithm Flow for Parameters

The algorithm flow for parameters solving $a_i$ is as follows:

**Step 1:** Take $n = 1$ as the initial value of the order *n* and suppose that there exists a small positive number $\varepsilon$.

**Step 2:** Take data points $\{v_i, p(v_i)\}$ from the measured wind speed and the responding distribution probability.

**Step 3:** By solving the equation set (13) using the least-squares method $a_i$ is acquired.

**Step 4:** Substitute $a_i$ into $p(v) = C \exp\left(\sum_{i=1}^{n} a_i v^i\right)$, and calculate the performance index *J*; if $J > \varepsilon$, renew the value of *n* according to $n = n + 1$, and then return to **Step 3**. Otherwise, end the loop and the current *n* value is the order of the exponential polynomial that we need.

**Step 5**: Record $a_i$ and calculate the distribution model of wind speed frequency from (10).

## 4. Simulations

To validate the proposed distribution model, we conducted simulation experiments based on measured data with two different distributions: unimodal distribution and multimodal distribution.

By utilizing the linear-least-squares method, we sought an optimal solution to the problem, such as the optimal value of $a_i$, minimizing the performance index *J*, or optimizing the frequency distribution model of wind speed to approximate the actual frequency distribution.

(1) Unimodal wind speed distribution

The data were collected from a wind tower at the height of 80 m in a mountainous area at an altitude above 1000 m, in the central part of China, from January to December 2013. The anemometer recorded a set of data every 10 min, and there were 52,560 groups of data after correction. Following calculation the annual average wind speed was found to be 5.05 m/s. Here the frequency distribution

of wind speed was calculated from the measured data from July. Then the Weibull distribution model, Rayleigh distribution model, log-normal distribution model and the exponential polynomial distribution model proposed in this paper were used to fit the measured data.

Through simulations, the comparison between wind speed frequency distribution for each model and the measured distribution is shown in Figure 1.

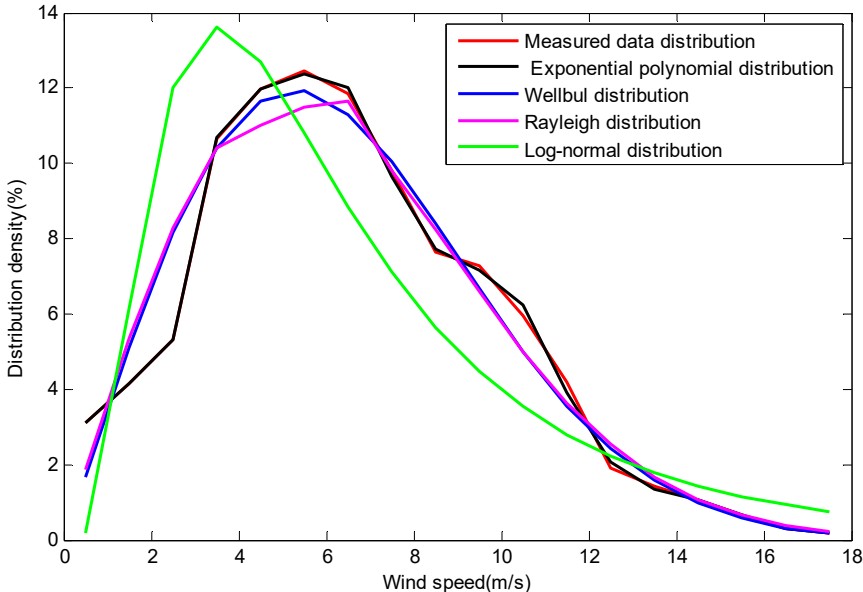

**Figure 1.** Comparison of frequency distribution of wind speed for each model in July.

The measured distribution probability and the calculation distribution probabilities of each model are shown in Table 1. The measured distribution density of wind speed is $f(v_i)/\%$, $f_W(v_i)/\%$ is the calculation density of the Weibull distribution model, $f_R(v_i)/\%$ is the calculation density of the Rayleigh distribution model, $f_R(v_i)/\%$ is the calculation density of the log-normal distribution model, and $f_E(v_i)/\%$ is the calculation density of the exponential polynomial distribution model.

**Table 1.** Comparisons of wind speed frequency between measured data and model calculations in July.

| Number $i$ | Wind Speed Section $v_i$ (m/s) | Mean Wind Speed $v_m$ (m/s) | $f(v_i)/\%$ | $f_W(v_i)/\%$ | $f_R(v_i)/\%$ | $f_N(v_i)/\%$ | $f_E(v_i)/\%$ |
|---|---|---|---|---|---|---|---|
| 1 | 0–1 | 0.5 | 3.11 | 1.68 | 1.86 | 0.20 | 3.11 |
| 2 | 1–2 | 1.5 | 4.14 | 5.13 | 5.36 | 6.14 | 4.14 |
| 3 | 2–3 | 2.5 | 5.33 | 8.16 | 8.30 | 12.02 | 5.32 |
| 4 | 3–4 | 3.5 | 10.66 | 10.40 | 10.39 | 13.61 | 10.69 |
| 5 | 4–5 | 4.5 | 11.98 | 11.66 | 11.50 | 12.68 | 11.97 |
| 6 | 5–6 | 5.5 | 12.46 | 11.92 | 11.67 | 10.83 | 12.38 |
| 7 | 6–7 | 6.5 | 11.85 | 11.30 | 11.03 | 8.86 | 12.03 |
| 8 | 7–8 | 7.5 | 9.79 | 10.04 | 9.80 | 7.11 | 9.65 |
| 9 | 8–9 | 8.5 | 7.66 | 8.4 | 8.24 | 5.65 | 7.72 |
| 10 | 9–11 | 9.5 | 7.28 | 6.66 | 6.58 | 4.47 | 7.17 |
| 11 | 10–11 | 10.5 | 5.96 | 5.00 | 5.01 | 3.54 | 6.23 |
| 12 | 11–12 | 11.5 | 4.21 | 3.57 | 3.64 | 2.80 | 3.91 |
| 13 | 12–13 | 12.5 | 1.90 | 2.43 | 2.53 | 2.23 | 2.05 |
| 14 | 13–14 | 13.5 | 1.41 | 1.57 | 1.68 | 1.78 | 1.34 |
| 15 | 14–15 | 14.5 | 1.05 | 0.97 | 1.07 | 1.43 | 1.076 |
| 16 | 15–16 | 15.5 | 0.67 | 0.57 | 0.66 | 1.15 | 0.67 |
| 17 | 16–17 | 16.5 | 0.31 | 0.32 | 0.38 | 0.93 | 0.31 |
| 18 | 17–18 | 17.5 | 0.20 | 0.17 | 0.22 | 0.76 | 0.20 |

The fitting error of each model for different wind speed section is calculated using the formula:

$$e(v_i) = |p_m(v_i) - p(v_i)| \tag{14}$$

The calculated results are given in Table 2.

**Table 2.** Comparisons of the fitting error among four models in July.

| Number $i$ | Wind Speed Section $v_i$ (m/s) | $e_W(v_i)$/% | $e_R(v_i)$/% | $e_N(v_i)$/% | $e_E(v_i)$/% |
|---|---|---|---|---|---|
| 1 | 0–1 | 1.43 | 1.25 | 2.91 | 0 |
| 2 | 1–2 | 0.99 | 1.22 | 2.00 | 0 |
| 3 | 2–3 | 2.83 | 2.97 | 6.69 | 0.01 |
| 4 | 3–4 | 0.26 | 0.27 | 2.95 | 0.03 |
| 5 | 4–5 | 0.32 | 0.95 | 0.70 | 0.01 |
| 6 | 5–6 | 0.54 | 0.96 | 1.63 | 0.08 |
| 7 | 6–7 | 0.55 | 0.18 | 2.99 | 0.18 |
| 8 | 7–8 | 0.25 | 0.01 | 2.68 | 0.14 |
| 9 | 8–9 | 0.74 | 0.58 | 2.01 | 0.06 |
| 10 | 9–11 | 0.62 | 0.70 | 2.81 | 0.11 |
| 11 | 10–11 | 0.96 | 0.95 | 2.42 | 0.27 |
| 12 | 11–12 | 0.64 | 0.57 | 1.41 | 0.30 |
| 13 | 12–13 | 0.53 | 0.63 | 0.33 | 0.15 |
| 14 | 13–14 | 0.16 | 0.27 | 0.37 | 0.07 |
| 15 | 14–15 | 0.08 | 0.02 | 0.38 | 0.03 |
| 16 | 15–16 | 0.10 | 0.01 | 0.48 | 0 |
| 17 | 16–17 | 0.01 | 0.07 | 0.62 | 0 |
| 18 | 17–18 | 0.03 | 0.02 | 0.56 | 0 |

Figure 2 provides the results of various $n$ values: $n = 5$, $n = 9$, $n = 13$. We know that as the order of the exponential polynomial increases, the error between $p_m(v_i)$ and $p(v_i)$ becomes smaller and smaller, and the fitting result also becomes more accurate.

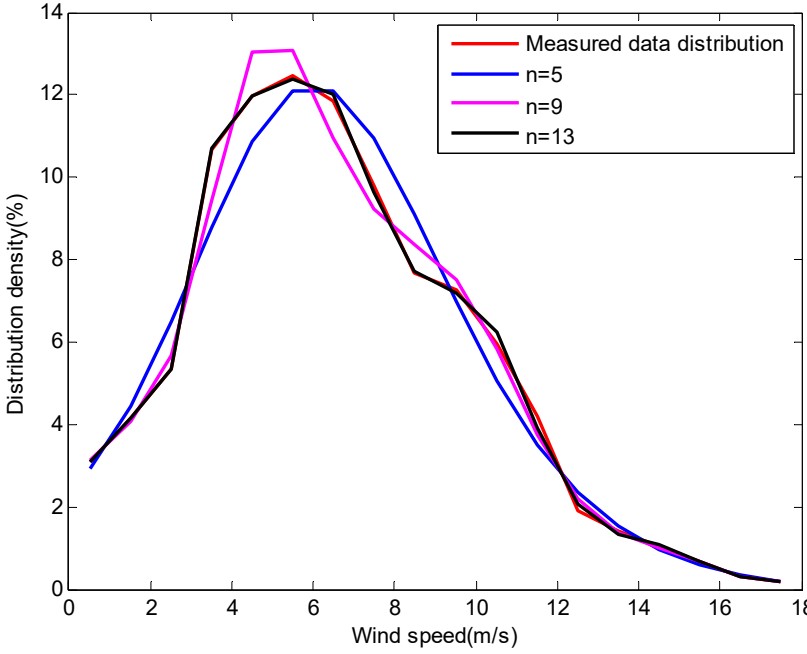

**Figure 2.** Comparison results when $n = 5$, $n = 9$, $n = 13$ for exponential polynomial distribution model.

When $n = 13$, the exponential polynomial distribution model fits the measured data very well, the normalised constant is $C = 0.0102$, and the parameters in Equation (10) are respectively:

$$
\begin{cases}
a_0 = -6.5762 \\
a_1 = 49.5920 \\
a_2 = -71.3296 \\
a_3 = 52.8481 \\
a_4 = -23.3786 \\
a_5 = 6.7090 \\
a_6 = -1.3094 \\
a_7 = 0.1783 \\
a_8 = -0.01711 \\
a_9 = 0.0001151 \\
a_{10} = -5.312 \times 10^{-5} \\
a_{11} = 1.601 \times 10^{-6} \\
a_{12} = -2.837 \times 10^{-8} \\
a_{13} = 2.24 \times 10^{-10}
\end{cases} \tag{15}
$$

From (10), we obtain the exponential polynomials frequency distribution model of wind speed:

$$
\begin{aligned}
p(v) = {}& 0.0102\exp\big(-6.5762 + 49.5920v - 71.3296v^2 + 52.8481v^3 - 23.3786v^4 \\
& + 6.7090v^5 - 1.3094v^6 + 0.1783v^7 - 0.01711v^8 + 0.0001151v^9 \\
& - 5.312 \times 10^{-5}v^{10} + 1.601 \times 10^{-6}v^{11} - 2.837 \times 10^{-8}v^{12} + 2.24 \times 10^{-10}v^{13}\big)
\end{aligned} \tag{16}
$$

The measured data for each month is fitted with the Weibull distribution model, the Rayleigh distribution model, the log-normal distribution model and the exponential polynomial distribution model. Fitting error accuracy for each month over the whole year is calculated according to formula (11), namely the value of $J$, as shown in Table 3.

**Table 3.** Fitting error accuracy comparison of various models.

| Month | Mean Wind Speed $v_m$ (m/s) | $J_W$ | $J_R$ | $J_N$ | $J_E$ |
|---|---|---|---|---|---|
| 1 | 4.64 | 0.0015 | 0.0021 | 0.0103 | $3.2164 \times 10^{-5}$ |
| 2 | 5.43 | 0.0020 | 0.0027 | 0.0178 | $5.8530 \times 10^{-5}$ |
| 3 | 5.58 | 0.0011 | 0.0014 | 0.0096 | $1.6742 \times 10^{-5}$ |
| 4 | 5.34 | 0.0018 | 0.0021 | 0.0119 | $4.7190 \times 10^{-5}$ |
| 5 | 4.07 | 0.0013 | 0.0021 | 0.0098 | $2.0962 \times 10^{-5}$ |
| 6 | 4.42 | 0.0015 | 0.0017 | 0.0106 | $2.6678 \times 10^{-5}$ |
| 7 | 6.49 | 0.0014 | 0.0016 | 0.0106 | $2.6640 \times 10^{-5}$ |
| 8 | 5.64 | 0.0011 | 0.0025 | 0.0102 | $1.4310 \times 10^{-5}$ |
| 9 | 5.35 | 0.0019 | 0.0017 | 0.02430 | $2.9308 \times 10^{-5}$ |
| 10 | 4.44 | 0.0025 | 0.0013 | 0.0212 | $3.6812 \times 10^{-5}$ |
| 11 | 4.11 | 0.0013 | 0.0100 | 0.0055 | $1.8920 \times 10^{-5}$ |
| 12 | 4.80 | 0.0013 | 0.0030 | 0.0088 | $2.0639 \times 10^{-5}$ |

In Table 3, $J_W$, $J_R$, $J_N$ and $J_E$, respectively, represent the fitting error accuracy of the Weibull distribution model, the Rayleigh distribution model, the log-normal distribution model and the exponential polynomial distribution model.

(2) Multimodal wind speed distribution

To further verify the effectiveness of the exponential polynomial model, we compared it to a study [20] where splines were used as wind speed frequency distribution functions, mainly for multi-modal wind speed distribution. From the simulation results, the model proposed in the literature

can adequately fit the measured data, whether for unimodal or multimodal wind speed distribution. But the disadvantage of this model is that it is limited by many constraints and needs to be initialised when calculating the parameters of the model. Here, we carried out fitting experiments based on the measured data of multimodal wind speed distribution using the exponential polynomial model. The simulation results are shown in Figure 3.

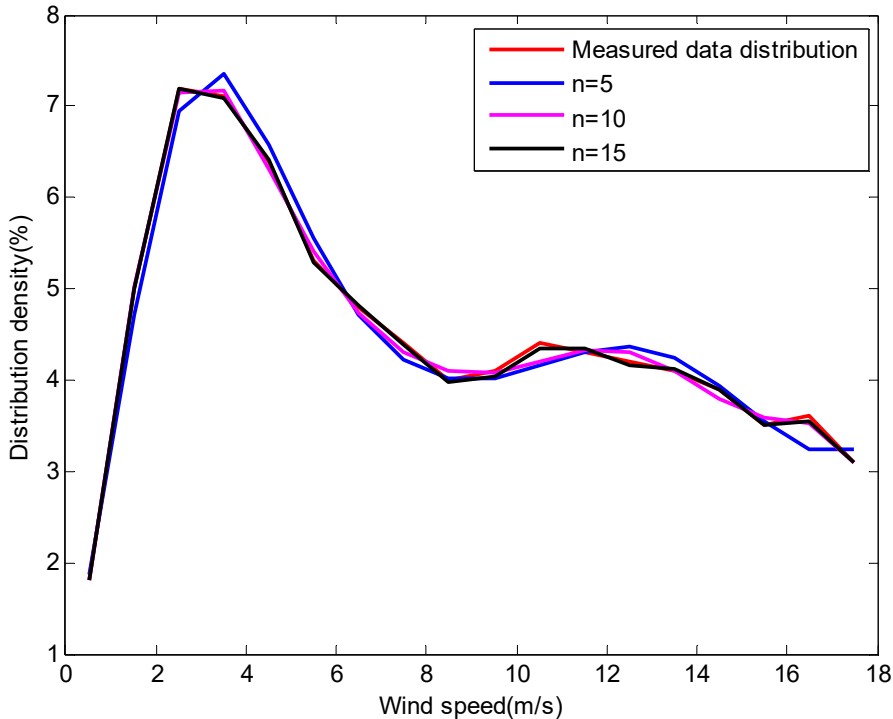

**Figure 3.** Comparison results when $n = 5$, $n = 10$, $n = 15$ for multimodal.

When $n = 15$, the normalised constant is $C = 0.0127$, and the parameters in Equation (10) are listed as follows:

$$
\begin{cases}
a_0 = 21.91 \\
a_1 = -76.5 \\
a_2 = 129.6 \\
a_3 = -115.2 \\
a_4 = 63.18 \\
a_5 = -23.17 \\
a_6 = 5.963 \\
a_7 = -1.108 \\
a_8 = 0.1513 \\
a_9 = -0.01527 \\
a_{10} = 0.001137 \\
a_{11} = -6.157 \times 10^{-5} \\
a_{12} = 2.359 \times 10^{-6} \\
a_{13} = -6.058 \times 10^{-8} \\
a_{14} = 9.351 \times 10^{-10} \\
a_{15} = -6.558 \times 10^{-12}
\end{cases}
\tag{17}
$$

Substituting $a_i$ into $p(v) = C \exp\left(\sum\limits_{i=1}^{n} a_i v^i\right)$, the exponential polynomial frequency distribution model of wind speed can be obtained:

$$
\begin{aligned}
p(v) =\ & 0.0102 \exp(21.91 - 76.5v + 129.6v^2 - 115.2v^3 + 63.186v^4 \\
& -23.17v^5 + 5.963v^6 - 1.1083v^7 + 0.1513v^8 - 0.01527v^9 \\
& +0.001137v^{10} - 6.157 \times 10^{-5}v^{11} + 2.359 \times 10^{-6}v^{12} - 6.558 \times 10^{-8}v^{13} \\
& +9.351 \times 10^{-10}v^{14} - 6.558 \times 10^{-12}v^{15})
\end{aligned}
\tag{18}
$$

## 5. Results Analysis

From Figure 1 in the simulation results, the fitting effect of the log-normal distribution model is the worst of the four distribution models. The Weibull distribution and Rayleigh distribution models are better than the log-normal distribution model. The fitting of the two models is somewhat close, especially at low and high wind speed, indicating that Rayleigh distribution is a special case of Weibull distribution. The exponential polynomial distribution model is prominently the best in fitting effect, and its absolute advantage lies in the excellent fitting at low wind speed, while the other three models all have a big gap between measured distribution and the distribution model. In addition, the fitting result of the exponential polynomial distribution model is also better than the other models at a high wind speed in Figure 1.

From Tables 1 and 2, it is apparent that the calculated probability of the exponential polynomial distribution model is closest to the measured probability overall, and the responding error is also smallest among the four distribution models. Encouragingly, for the low wind speed section of 0–1 m/s and 1–2 m/s in Table 2, the error between the calculated wind speed probability of the exponential polynomial model and the measured wind speed probability is 0, which fully illustrates the outstanding advantage of the proposed model. Contrarily, the calculated probability for the log-normal distribution model is farthest from the measured probability, and the error is largest. With regards to the Weibull and Rayleigh distribution models, the calculated probability of the former is relatively closer to the measured probability than that of the latter, and this is also demonstrated by the error in Table 2, which is the reason that Weibull distribution is often considered to be a better model for describing wind speed frequency distribution in much of the literature, except for the problems with zero and low wind speed. A more important aspect to consider for wind farm design is that the error is low when the potential power production is high, which is reflected by the 0 error for the high wind speed section of 15–16 m/s, 16–17 m/s and 17–18 m/s in Table 2.

Table 3 provides the fitting error accuracy of the four models for annual wind speed. The error accuracy of the distribution model proposed in this paper is far less than that of the other three models, with the error accuracy reaching $10^{-5}$. This indicates that the fitting effect is the best. The fitting error accuracy of the Weibull distribution model is the second best, and that of the log-normal distribution model is the worst. So, among the four models the exponential polynomial distribution model is the most suitable model for wind speed frequency distribution. At the same time, we noticed that in September and October the fitting of the Weibull distribution model to the measured data was worse than that of the Rayleigh distribution model, which is probably due to the variable wind direction and instability of mountain winds during these two months.

Figure 3 shows that the exponential polynomial distribution model can also fit the measured wind speed data with multimodal distribution. In order to achieve even better fitting results the order of the exponential polynomial model can be set much higher. This is an advantage that other models cannot surpass, especially for the case of multimodal wind speed distribution.

## 6. Conclusions

In this paper we put forward an exponential polynomial distribution model to describe and calculate the frequency distribution of wind speed. The proposed distribution model not only solves

the problem that the probability density is not zero at zero wind speed, but also improves the problem of a big gap between model calculation and measured data at low wind speed. This can reproduce the non-vanishing probability of 0 or almost 0 wind speed much better, which is useful in wind farm design, because one can estimate the hours of wind below the wind turbine cut-in. At the same time, the distribution model has smaller errors at high wind speed, which is much more significant for the higher potential power production.

Moreover, the exponential polynomial distribution model can fit multimodal distribution wind speed data as well as unimodal distribution wind speed data. With an increase to the order of the exponential polynomial, the fitting effect is correspondingly improved. Although the number of parameters required is large for the best fitting effect, it is very convenient to calculate by adopting the linear-least-squares method.

To further improve the practicability of the model, future work includes tests on different data sets, such as different time periods for the same location or investigating different locations.

**Author Contributions:** L.W. conceived the work and performed the experiments, drafted and revised the manuscript; J.L. helped to make an analysis with constructive discussions; F.Q. approved the final version.

**Funding:** This research was funded by [National Natural Science Foundation of China] grant number [61773016, 61473222, 61873201], and [Scientific Research Plan Projects of Shannxi Education Department] grant number [16JK1690].

**Conflicts of Interest:** The authors declare that they have no conflict of interest.

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
