# Peer review of "Frequency Distribution Model of Wind Speed Based on the Exponential Polynomial for Wind Farms"

_sustainability, doi:10.3390/su11030665_

Round 1

Reviewer 1 Report

This paper experienced a significant improvement.

Some of my concerns were addressed.

However, a few things should be improved.

First, the English has to be thoroughly improved.An example is for instance: "Through comparison experiments, the proposed" line 344. 

A thorough proofreading of the article is necessary regarding English.

Why are the equation terms on lines 149, 177, 178, 194, 195 so stretched? Why can't the authors just make them proportional?

In Figure 1, the legend box is obstructing the lines (the results). Please move it aside so the entire plot could be visible. Also, why is the legend "measured data distribution" with lower case? Please capitalize all the legend box.

The conclusion is extensive yet incomplete. The obtained results should be briefly and clearly mentioned through the support of numerical data. What were the most sounding quantifiable findings of this study? The conclusion is intended to help the reader understand why your research should matter to them after they have finished reading the paper.

The introduction section could include a broader picture of the "wind energy resources" in the research community. Thus, the following articles could be added to the introduction section:

   https://doi.org/10.3390/su10103811

   https://doi.org/10.1109/TIA.2017.2754978

   https://doi.org/10.1109/TSTE.2015.2455552

   https://doi.org/10.1109/TSTE.2018.2789398

The authors have to clearly improve the article.

Author Response

Response to Reviewer 1 Comments

Point 1: First, the English has to be thoroughly improved.An example is for instance: "Through comparison experiments, the proposed" line 344. A thorough proofreading of the article is necessary regarding English.

Response 1: Thank you very much for pointing out our incorrect expression, and we have made revision for it. At the same time, we have polished the English using a professional English editing service for service extensive English editing.

Point 2: Why are the equation terms on lines 149, 177, 178, 194, 195 so stretched? Why can't the authors just make them proportional?

Response 2: We have adjusted the equation terms on lines 149, 177, 178, 194, 195 and made them proportional.

Point 3: In Figure 1, the legend box is obstructing the lines (the results). Please move it aside so the entire plot could be visible. Also, why is the legend "measured data distribution" with lower case? Please capitalize all the legend box.

Response 3: Thank you for your approvals, and we have revised them.

Figure 1. Comparison of frequency distribution of wind speed for each model in July

Figure 2. Comparison results when ,,

for exponential polynomial distribution model

Figure 3. Comparison results when ,, for multimodal

wind speed based on exponential polynomial distribution model

Point 4: The conclusion is extensive yet incomplete. The obtained results should be briefly and clearly mentioned through the support of numerical data. What were the most sounding quantifiable findings of this study? The conclusion is intended to help the reader understand why your research should matter to them after they have finished reading the paper.

Response 4: We have supplemented the conclusions.

Conclusions

In this paper we put forward an exponential polynomial distribution model to describe and calculate the frequency distribution of wind speed. The proposed distribution model not only solves the problem that the probability density is not zero at zero wind speed, but also improves the problem of a big gap between model calculation and measured data at low wind speed. This can much better reproduce the non-vanishing probability of 0 or almost 0 wind speed, which is useful in wind farm design, because one can estimate the hours of wind below the wind turbine cut-in. At the same time, the distribution model has smaller errors at high wind speed, which is much more significant for the higher potential power production.

Moreover, the exponential polynomial distribution model can fit multimodal distribution wind speed data as well as unimodal distribution wind speed data. With an increase to the order of the exponential polynomial, the fitting effect is correspondingly improved. Although the number of parameters required is large for the best fitting effect, it is very convenient to calculate by adopting the linear-least-squares method.

To further improve the practicability of the model, future work includes tests on different data sets, such as different time periods for the same location or investigating different locations.    

Point 5: The introduction section could include a broader picture of the "wind energy resources" in the research community. Thus, the following articles could be added to the introduction section:

   https://doi.org/10.3390/su10103811

   https://doi.org/10.1109/TIA.2017.2754978

   https://doi.org/10.1109/TSTE.2015.2455552

   https://doi.org/10.1109/TSTE.2018.2789398

Response 5: Thank you for your approval and offering the articles. We have added them in the introduction section.

Investment in renewable energy sources, including wind power plants, is of particular importance because of the increased efficiency of clean energy, and the need to reduce pollution and fuel consumption [1]. As wind generation technologies improve, this form of energy production becomes a valuable alternative to conventional energy sources [2]. The proportion of energy generated by wind is increasing due to recent technology and efficiency improvements, as well as government funding [3]. An important problem in using wind power is their uncertain nature and the characteristic of being unforeseen [4].

1.   Jaber, V.; Mousa, M.; Mudathir, F.A.; Ian, D.E.; Radu, G.; JoĂŁo Carlos de Oliveira Matias; Edris, P. Long-Term Decision on Wind Investment with Considering Different Load Ranges of Power Plant for Sustainable Electricity Energy Market. Sustainability, 2018, 10(10), 3811; doi:10.3390/su10103811.

2.  Shi, J.; Lee, W.J; Liu, X.F. Generation Scheduling Optimization of Wind-Energy Storage System Based on Wind Power Output Fluctuation Features. IEEE Transactions on Industry Applications, 2018, 54, (1), 10-17.

3.   Mojgan, H.M.; Zhang, J.S.; Kory, W.H. Wind Power Dispatch Margin for Flexible Energy and Reserve Scheduling With Increased Wind Generation. IEEE Transactions on Sustainable Energy, 2015, 6(4), 1543 – 1552.

4.  Arash, B.; Jamshid, A.; Taher, N.; Miadreza, S.K.; Radu, G.; Joao, P.S.CatalĂŁo. Bundled Generation and Transmission Planning Under Demand and Wind Generation Uncertainty Based on a Combination of Robust and Stochastic Optimization. IEEE Transactions on Sustainable Energy, 2018, 9(3) ,1477-1486.

Reviewer 2 Report

The manuscript has substantially improved with respect to the first version that was rejected.

In particular, the authors put better their study in the context about the subject and they use their methods also for a comparison against a study in the literature: this is a reasonable support to the fact that their results can be general or generalizable.

I have one further recommendation: the focus of the work is in the fact that the proposed method can much better reproduce the non-vanishing probability of 0 or almost 0 wind intensity. This is useful in the context of wind farm design, because one can estimate the hours of wind below the wind turbine cut-in: in my opinion this should said more clearly (more or less like I have said it here). A further aspect is that for wind farm design is even more important that the error is low when the potential power production is higher: watching your results, the distributions you propose look better also when the wind speed is high. This point, in my opinion, should as well be emphasized.  

Author Response

Response to Reviewer 2 Comments

Point 1: I have one further recommendation: the focus of the work is in the fact that the proposed method can much better reproduce the non-vanishing probability of 0 or almost 0 wind intensity. This is useful in the context of wind farm design, because one can estimate the hours of wind below the wind turbine cut-in: in my opinion this should said more clearly (more or less like I have said it here). A further aspect is that for wind farm design is even more important that the error is low when the potential power production is higher: watching your results, the distributions you propose look better also when the wind speed is high. This point, in my opinion, should as well be emphasized. 

Response 1: Thank you very much for your kind advice, and we have added your recommendations to the results analysis and conclusions.

4. Results Analysis

From Fig. 1 in the simulation results, the fitting effect of the log-normal distribution model is the worst of the four distribution models. The Weibull distribution and Rayleigh distribution models are better than the log-normal distribution model. The fitting of the two models is somewhat close, especially at low and high wind speed, indicating that Rayleigh distribution is a special case of Weibull distribution. The exponential polynomial distribution model is prominently the best in fitting effect, and its absolute advantage lies in the excellent fitting at low wind speed, while the other three models all have a big gap between measured distribution and the distribution model. In addition, the fitting result of the exponential polynomial distribution model is also better than the other models at high wind speed in Fig. 1.

From Tables 1 and 2, it is apparent that the calculated probability of the exponential polynomial distribution model is closest to the measured probability overall, and the responding error is also smallest among the four distribution models. Encouragingly, for the low wind speed section of 0-1 m/s and 1-2 m/s in Table 2, the error between the calculated wind speed probability of the exponential polynomial model and the measured wind speed probability is 0, which fully illustrates the outstanding advantage of the proposed model. Contrarily, the calculated probability for the log-normal distribution model is farthest from the measured probability, and the error is largest. With regards to the Weibull and Rayleigh distribution models, the calculated probability of the former is relatively closer to the measured probability than that of the latter, and this is also demonstrated by the error in Table 2, which is the reason that Weibull distribution is often considered to be a better model for describing wind speed frequency distribution in much of the literature, except for the problems with zero and low wind speed. A more important aspect to consider for wind farm design is that the error is low when the potential power production is high, which is reflected by the 0 error for the high wind speed section of 15-16 m/s, 16-17 m/s and 17-18 m/s in Table 2.

Table 3 provides the fitting error accuracy of the four models for annual wind speed. The error accuracy of the distribution model proposed in this paper is far less than that of the other three models, with the error accuracy reaching 10-5. This indicates that the fitting effect is the best. The fitting error accuracy of the Weibull distribution model is the second best, and that of the log-normal distribution model is the worst. So, among the four models the exponential polynomial distribution model is the most suitable model for wind speed frequency distribution. At the same time, we noticed that in September and October the fitting of the Weibull distribution model to the measured data was worse than that of the Rayleigh distribution model, which is probably due to the variable wind direction and instability of mountain winds during these two months.

Fig. 3 shows that the exponential polynomial distribution model can also fit the measured wind speed data with multimodal distribution. In order to achieve even better fitting results the order of the exponential polynomial model can be set much higher. This is an advantage that other models cannot surpass, especially for the case of multimodal wind speed distribution.

5. Conclusions

In this paper we put forward an exponential polynomial distribution model to describe and calculate the frequency distribution of wind speed. The proposed distribution model not only solves the problem that the probability density is not zero at zero wind speed, but also improves the problem of a big gap between model calculation and measured data at low wind speed. This can much better reproduce the non-vanishing probability of 0 or almost 0 wind speed, which is useful in wind farm design, because one can estimate the hours of wind below the wind turbine cut-in. At the same time, the distribution model has smaller errors at high wind speed, which is much more significant for the higher potential power production.

Moreover, the exponential polynomial distribution model can fit multimodal distribution wind speed data as well as unimodal distribution wind speed data. With an increase to the order of the exponential polynomial, the fitting effect is correspondingly improved. Although the number of parameters required is large for the best fitting effect, it is very convenient to calculate by adopting the linear-least-squares method.

To further improve the practicability of the model, future work includes tests on different data sets, such as different time periods for the same location or investigating different locations.

Round 2

Reviewer 1 Report

The authors have addressed all the concerns of the reviewer.

This manuscript is a resubmission of an earlier submission. The following is a list of the peer review reports and author responses from that submission.

Round 1

Reviewer 1 Report

The manuscript entitled “Frequency distribution model of wind speed based on exponential polynomial for wind farms” in my opinion doesn’t have the sufficient quality for being considerable acceptable for publication.

The literature review is poor: the authors often refer to old studies. They should instead base the motivations of their work on a fertile discussion with the up-to-date state of the art in the literature.

The rationale for their wind speed distribution proposal is motivated somehow (the discrepancy between wind speed distributions that are common in the literature and the measurements in the very low wind intensity tail), but the physical sense of the selected model type is not explained at all. Why exponential polynomials and not something else? Is there a reason?

Only one data set of one year is used for validating the proposal and the authors claim the superiority of their proposed wind speed distribution only on this basis. In my opinion, it is definitely not sufficient. These results might likely be a chance, depending on the features of the particular data set. Furthermore, the discussion of the results is poor. The authors could take inspiration from this study, that is much more rigorous: Dookie, I., Rocke, S., Singh, A., & Ramlal, C. J. (2018). Evaluating wind speed probability distribution models with a novel goodness of fit metric: a Trinidad and Tobago case study. International Journal of Energy and Environmental Engineering, 1-17.

On these grounds, my opinion is that the manuscript in its present form should be rejected

Reviewer 2 Report

1.     The authors have made an interesting attempt to write a paper addressing a frequency distribution model of wind speed based on exponential polynomial for wind farms. However, the paper is short and lacks depth.

2.     Introduction: From my point of view, introduction is not well focused. In a research paper, it is expected that introduction section briefly explains the starting background and, even more important, the originality (novelty) and relevancy of the study is well established. Once this is done, hypothesis and objectives of the study need to be addressed, as well as a brief justification of the conducted methodology. It is my belief that, in this case, authors do not put effort enough (or any effort) in highlighting the relevancy and (specially) the novelty of the study. Consequently, both major aspects are compromised. I strongly recommend that authors clearly explain all these aspects (including hypothesis and objectives) in order to add scientific rigor to the manuscript.

3.     Discussion: It is my opinion that a separate discussion section would help the reader to understand the study. However, the main issue arises from the lack of comparison with state-of-the-art studies. From my point of view, even though many publications are listed in the literature review, it has no justification not conducting a comparison with main previous works already published. The lack of such a comparison highly compromises the significance of the paper, so I strongly recommend authors to conduct as much and suitable comparisons as needed to solve this issue.

4.     Why is the Introduction chapter 0?

5.     The paper is too short and lacks depth.

6.     Section 3 of the results analysis should be much more developed. It is too short as well and more attention to detail should be given.

7.     The quality of presentation of this paper is very low.

8.     What is wrong with the symbols in the lines 129 – 130?

9.     This paper is full of typos showing no attention to detail. For instance what is “CALIF1 [9]” on line 48? Or what is reference 9 repeated twice on line 283?

10.  Why are equation of lines 219 and 222 not numbered?

11.  This paper is full of typos which shows a lack of attention to detail.

12.  The conclusion is extensive yet incomplete. The obtained results should be briefly and clearly mentioned through the support of numerical data. What were the most sounding quantifiable findings of this study? The conclusion is intended to help the reader understand why your research should matter to them after they have finished reading the paper. A conclusion is not merely a summary of the main topics covered or a re-statement of your research problem, but a synthesis of key points and, if applicable, where you recommend new areas for future research.

13.  This paper seems to be adequate for conference proceedings and for such type of papers is acceptable since this paper shows many limitations. However, the authors need much to improve in order to have an acceptable paper for Sustainability.